# TOWARDS ONLINE REAL-TIME MEMORY-BASED VIDEO INPAINTING TRANSFORMERS

## ABSTRACT

Video inpainting tasks have seen significant improvements in the past years with the rise of deep neural networks and, in particular, vision transformers. Although these models show promising reconstruction quality and temporal consistency, they are still unsuitable for live videos, one of the last steps to make them completely convincing and usable. The main limitations are that these state-of-the-art models inpaint using the whole video (offline processing) and show an insufficient frame rate. In our approach, we propose a framework to adapt existing inpainting transformers to these constraints by memorizing and refining redundant computations while maintaining a decent inpainting quality. Using this framework with some of the most recent inpainting models, we show great online results with a consistent throughput above 20 frames per second. Code and pretrained models will be made available upon acceptance.

## 1 INTRODUCTION

Video inpainting is the task of filling missing regions in a video with plausible and coherent content. It can be seen as an extension of the more known image inpainting but with the extra temporal dimension, bringing new challenges. A good video inpainting can be used for various applications, such as object removal (Ebdelli et al., 2015), video restoration (Lee et al., 2019) or video completion (Chang et al., 2019b).

To be convincing, a video inpainting must be spatially coherent, that is, the content filled in each frame fits with the rest of the image. It must also be temporally coherent, meaning that the video is smooth and without artifacts when being played. Models leveraging a deep learning approach (Kim et al., 2019; Oh et al., 2019; Chang et al., 2019a) have made significant progress recently, especially on the temporal consistency that was lacking from more traditional methods (Wexler et al., 2007; Granados et al., 2012; Newson et al., 2014). Among them, the transformers (Zeng et al., 2020; Liu et al., 2021a) showed the best performance both in terms of quality and speed.

With more and more live content today (e.g. cultural and sport events, social media streaming), online and real-time video inpainting is necessary to deal with these new types of broadcast. Such techniques could also prove to be useful in the augmented perception field. These models should be able to inpaint an ongoing video, with sufficient frame rate to be 'live'. While a few previous works investigated this (Herling & Broll, 2014; Kari et al., 2021), none of the current state-of-the-art approaches meet the criteria to be called either online or real-time, limiting the potential real-life use cases of this technology.

In this work, we propose a framework to adapt the most recent transformer-based techniques of video inpainting to both online and real-time standards, with as little loss of quality as possible.

**I. Online** We explore the natural modifications to make any inpainting model work *online*. By doing that, we derive an online baseline trading off inpainting quality. The main drawback of this approach is the frame rate, which is still too low for real time.

**II. Memory** We then add a *memory* and keep the successive results of these transformers, to reduce the number of calculations to do for the next frames. With that, we increase the number of frames per second by a factor of 3, passing the real-time threshold at the cost of yet another quality drop.

**III. Refined**    Finally, we *refine* the memory-based framework to temper the loss of inpainting quality while maintaining a real-time throughput. To do that, two models run side by side and communicate together as the live video goes on. The first one inpaints the frames in real-time as they come, using as much previous knowledge as it can have. Simultaneously, the second model reinpaints already gone frames with more time and more care. It then communicates its results to the first model, giving it valuable information to use.

We demonstrate the proposed techniques (*Online, Memory, Refined*) on three of the most recent transformer-based models and achieve online real-time operating points when testing on the usual video inpainting tasks and datasets.

The remainder of the paper is structured as follows. In Section 2, we give an overview of the former and current research on video inpainting. Then, we detail our online video inpainting models in Section 3. Finally, we report all our results and discuss them in Section 4.

## 2 RELATED WORK

### 2.1 IMAGE INPAINTING

The first works on image inpainting were proposed decades ago and were relying on the use of known textures to fill the missing content (Bertalmio et al., 2000; Efros & Leung, 1999; Efros & Freeman, 2001). These textures were sampled from other areas of the image thanks to patches that were matched with the corrupted area with a similarity score. Variations around this approach have then been proposed (Hays & Efros, 2007) including PatchMatch (Barnes et al., 2009), using an approximation of the patch matching to obtain a tool fast enough for commercial uses.

With the development of more complex models such as convolutional neural networks (CNN) (Krizhevsky et al., 2012), recurrent neural networks (RNN) (He et al., 2016) or generative adversarial networks (GAN) (Goodfellow et al., 2014), recent image inpainting works have focused on the use of deep neural networks (e.g. encoders) trained with adversarial losses, with great results (Iizuka et al., 2017; Pathak et al., 2016; Yu et al., 2018; Nazeri et al., 2019).

### 2.2 TRADITIONAL VIDEO INPAINTING

As for image inpainting, the first models proposed were handcrafted, relying on more traditional image manipulation techniques. Most models adopted a similar patch-based approach in which the video was cut into spatial-temporal patches. A score was also computed between the patches to fill the missing content with information from similar patches (Wexler et al., 2004; 2007; Newson et al., 2014; Patwardhan et al., 2005). To later deal with videos having more complex movements, refinements have been proposed, for example, introducing the use of flows to help the patch-based inpainting (Huang et al., 2016). Other techniques not using patches were also proposed, using, for instance, image transformations to align the frames together (Granados et al., 2012).

### 2.3 DEEP VIDEO INPAINTING

Deep learning brought significant improvement in the quality of video inpainting in the same way it did with image inpainting. Deep neural networks have been leveraged in various ways to inpaint a video (Ouyang et al., 2021; Lee et al., 2019; Ke et al., 2021), with most of them belonging to three main categories, as described by Zou et al. (2021).

One way to video inpaint is by employing an encoder-decoder model with a mix of 2D and 3D convolutions (Wang et al., 2019). VINet (Kim et al., 2019) was one of the first deep neural models capable of competing with state-of-the-art models using traditional techniques (Huang et al., 2016). Improvements were later made by adding gated convolutions (Chang et al., 2019a) and designing a video-specific GAN loss called T-PatchGAN (Chang et al., 2019b) used in many works since then (Zeng et al., 2020; Liu et al., 2021b). This strategy is, however, impeded by the heavy calculations that 3D convolutions bring.

Other models perform video inpainting by utilizing the optical flows of the videos (Xu et al., 2019; Gao et al., 2020). Forward and backward flows are first computed for the known part of the video

with traditional techniques such as FlowNet (Dosovitskiy et al., 2015; Ilg et al., 2017) before being completed by the models. Missing content is then filled by propagating known pixels through the obtained flows. This approach shows great results, especially at the border of the missing area, which is completed really smoothly. The main drawback of these models is the speed, with a throughput far from real-time because of the flow estimation and completion.

A final category encompasses the so-called attention-based models. While the earliest ones still leverage attention mechanisms in a traditional manner (Oh et al., 2019; Li et al., 2020), the most recent and powerful ones all rely on transformers (Vaswani et al., 2017; Dosovitskiy et al., 2021; Liu et al., 2021c) with different ways of splitting the frames into patches. While STTN (Zeng et al., 2020) uses patches of different sizes, one for each head of the attention mechanism, DSTT (Liu et al., 2021b) divides the transformers into two categories (spatial transformers and temporal transformers) to mix the patches differently. With FuseFormer (Liu et al., 2021a), the idea of having the patches overlapping to add more consistency gives even better results.

Finally, some works try to be more transverse and merge together different approaches. This is the case of TSAM (Zou et al., 2021) which combines 3D convolutions with flows. The current state-of-the-art work, E2FGVI (Li et al., 2022), is also following this approach by leveraging both FuseFormer-like filling with flows propagation.

## 2.4 ONLINE VIDEO INPAINTING

To our best knowledge, only two works seriously tried to tackle the challenge of online video inpainting. They both focus on object removal as part of an entire end-to-end framework also including object segmentation and tracking. The first one, PixMix (Herling & Broll, 2014), is more traditional and relies on pixel-wise homography mapping, which can only be used for specific movements of the camera and the object. The second one, TransforMR (Kari et al., 2021), is more recent and it adapts two of the early deep inpainting models VINet (Kim et al., 2019) and LGTSM (Chang et al., 2019a) without being able to get both quality and throughput simultaneously: one is fast but gives a quite poor result, the other is slow but inpaints better. Unfortunately, none of these two works actually provide quantitative results to compare with.

## 3 THE PROPOSED MODELS

### 3.1 PROBLEM FORMULATION AND EXISTING APPROACH

A video inpainting task is given by a set of consecutive corrupted frames $X := \{X_1, \ldots, X_N\}$ and corresponding masks $M := \{M_1, \ldots, M_N\}$ with $N$ the length of the video. Each pixel of a mask $M_i$ has either the value "0" if the true pixel is known or "1" if the value is missing. The goal of video inpainting is to use $X$ and $M$ to reconstruct the ground truth $Y := \{Y_1, \ldots, Y_N\}$. For the training, as $X$ and $Y$ are rarely known together (e.g., in object removal), $X$ is usually obtained in practice by corrupting a perfectly normal video $Y$ using an arbitrary mask $M$: $X = Y \odot (1 - M)$ with $\odot$ being an element-wise multiplication.

Recent attention-based techniques aim at completing a given frame by using the information in both short-term context (neighboring frames) and long-term context (reference frames). This enables a good temporal consistency while being able to find information further away in the video, which is for example useful when the mask is moving too slowly for the neighboring frames to be helpful. More specifically, all frames in a temporal window of radius $k$ centered around frame $f$ are completed simultaneously using all the frames of the window, on top of reference frames sampled in the whole video at a sampling rate $r$:

$$\hat{Y}_{f-k}^{f+k} = \mathcal{M}\left(X_{f-k}^{f+k} \cup X_{1,r}^{N}\right), \tag{1}$$

where $\hat{Y}_{f-k}^{f+k}$ and $X_{f-k}^{f+k}$ are the reconstructed and original frames in a window from frame $f - k$ to frame $f + k$, $X_{1,r}^{N}$ is the set of frames from the whole video sampled at rate $r$, and $\mathcal{M}$ is the inpainting model. An illustrated example is given in Figure 1 (a).

These frames are then encoded and cut into patches in different ways, depending on the model chosen (Liu et al., 2021a;b; Zeng et al., 2020). They are then completed using a self-attention

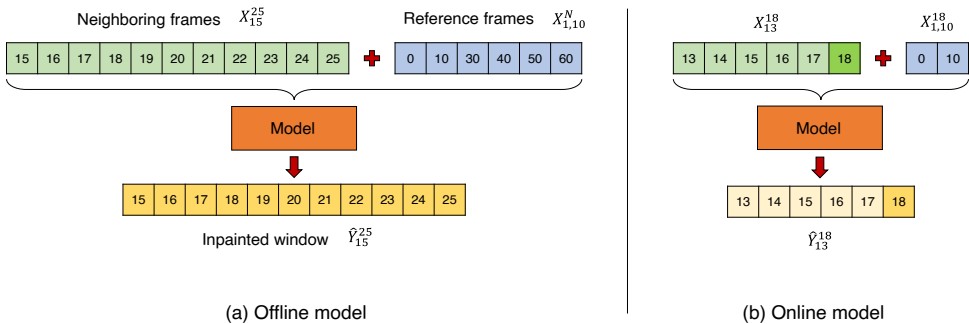

Figure 1: Original inpainting model and its natural online adaptation. **(a)** The model inpaints a window centered around $f = 20$ with a radius $k = 5$. Reference frames sampled at a rate $r = 10$ are added as input of the model. **(b)** In online inpainting, we can only see the past frames. To inpaint the frame 18, we use a window and sampled frames from the past. The whole window is still predicted but only the last frame is effectively used.

framework composed of 8 consecutive transformer blocks. The transformers follow the same general pipeline as the original vision transformer (Dosovitskiy et al., 2021) with slight modifications for each model. For instance, the final feed-forward layer is replaced by a custom Fusion-feed-forward layer in FuseFormer (Liu et al., 2021a). One model is also using an additional flow-based module (Li et al., 2022) to help the transformers completion.

## 3.2 CHALLENGES OF ONLINE INPAINTING

Inpainting a video in real-time raises three main issues regarding the current state-of-the-art models:

1. A frame must be inpainted using only information from the frames before because we cannot "see" in the future.

2. A new frame must be inpainted as soon as possible when received by the model, therefore we cannot wait for several frames to inpaint them together.

3. A sufficient throughput is required to be able to call that "real time". We fix the value of 20 frames per second (FPS) as our goal here.

## 3.3 ONLINE VIDEO INPAINTING TRANSFORMERS

Given these new constraints, a drop in the quality of video inpainting is expected and partly inevitable. For that reason, it is important to define a proper baseline for online inpainting so that we can fairly assess our approach. Comparing an online model with a regular all-seeing one would be unfair as they do not process the same information and are not meant for the same usage.

Taking into account the first two constraints, a straightforward baseline can be derived from any of the existing inpainting models. Frames are inpainted one by one, not anymore by blocks, and a given frame is inpainted only with information from the past. Using the same notations, this model is described in Equation equation 2 and Figure 1 (b).

$$\hat{Y}_f = \mathcal{M}\left(X_{f-k}^f \cup X_{1,r}^f\right). \tag{2}$$

While a drop in inpainting quality is expected, this model also shows a very poor frame rate (see Tables 1 and 2). This is explained by the fact that the model still inpaints a whole window of frames as in classical video inpainting, but only the prediction for the last frame is used here given constraints 1 and 2 (Figure 1). Starting from this baseline, we propose two models significantly increasing the frame rate while staying as close as possible to this one in terms of inpainting quality.

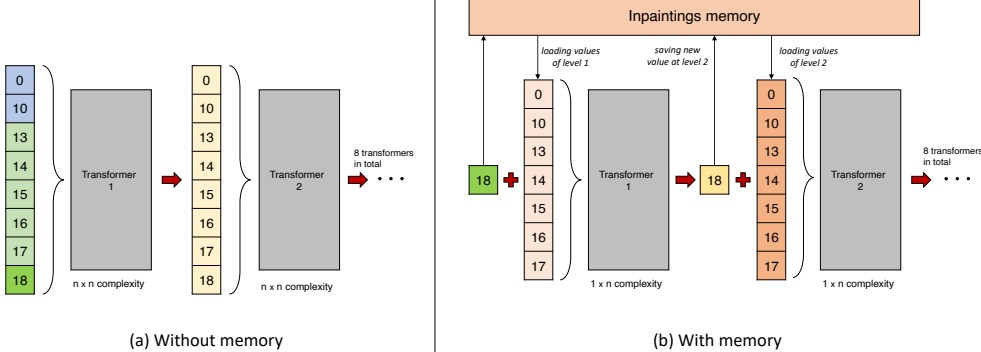

Figure 2: Transformers in the baseline and memory-based models. **(a)** Without memory, the baseline model processes all the frames in each transformer, making it quadratically complex. **(b)** When the memory of the previous inpaintings is kept, only the new frame (18) needs to be computed, while the transformers can still use the other frames (0 to 17) as context. After each transformer, the new result is saved for later. Each frame is saved in the memory as much times as there are transformers, each value being different. Following Equation equation 3, we have here $f = 18$, $s = 5$ and $r = 10$.

### 3.4 MEMORY-BASED ONLINE VIDEO INPAINTING TRANSFORMERS

This model addresses the frame rate problem by narrowing the prediction pipeline to only one frame, the last one available at a given time. The query vector in the attention mechanism is changed to only contain patches from this last frame, avoiding any unnecessary prediction of the other frames. Meanwhile, the key and value vectors must still represent all the frames selected for the prediction (neighboring and reference frames) to keep a good inpainting. This raises a problem as the original inpainting model is composed of several chained transformers (e.g., 8 transformers in Fuseformer or STTN), in which the results from the first one serve as input for the second one to construct the query, key and value vectors, and so on with the next ones (Figure 2 (a)).

To deal with that, we propose to save in memory the successive outputs of these transformers for each predicted frame and reuse them later for the next prediction. This approach effectively transforms a quadratic attention computation (n frames predict n frames) into a linear one (n frames predict 1 frame) as shown in Figure 2 (b). Moreover, the rest of the prediction pipeline now forwards one frame only as the other frames are only used by the transformers. This enables a gain of time on other operations such as convolutional encoding or patch splitting, which is not negligible in a model like Fuseformer using overlapping patches. The resulting equation is:

$$\hat{Y}_f = \mathcal{M}'\left(X_f, M_{f-s}^{f-1} \cup M_{1,r}^{f-1}\right), \tag{3}$$

with $\mathcal{M}'$ the new model, $M$ the transformers memory. $s$ and $r$ are the memory span and sampling rate: frames from index $f - s$ to $f - 1$, as well as previous frames sampled at rate $r$ will be used to inpaint frame $f$. A visual example can be found in Figures 2 (b) and 3 (a).

As detailed in Tables 1 and 2, this approach greatly improves the frame rate, leading to values admissible as "real time", i.e. above 20 FPS. The main drawback of this model is the significant reduction in video quality, as measured on all 4 metrics.

### 3.5 REFINED MEMORY-BASED ONLINE VIDEO INPAINTING TRANSFORMERS

Investigations made on the previous model show that if a frame is poorly inpainted, the results will still be saved in memory, never recomputed, and reused later for the following frames, maintaining a poor inpainting through the whole video. This is even worsened by the fact that the inpainting results are created independently, as only one frame is predicted at a time. Such a problem doesn't happen in the baseline model because a) frames are inpainted together in windows and not independently and b) a poor inpainting result for a given frame will not be used for the following inpaintings.

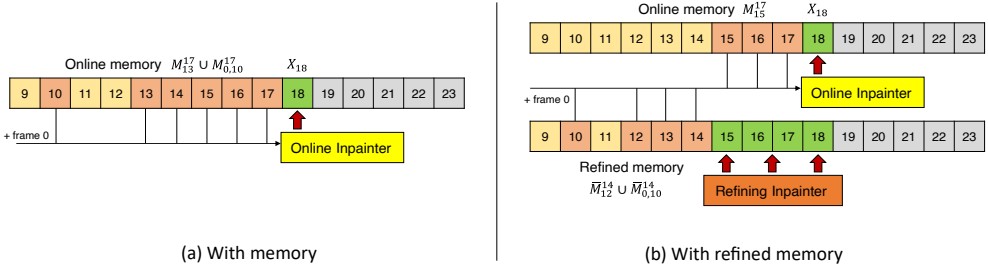

(a) With memory   (b) With refined memory

Figure 3: Memory-based and refined models. **(a)** Thanks to inpainting memory of the last seen frames, the new frame is the only one to be computed but the inpainting still benefits from this previous context. Following Equation equation 3, we have here $f = 18$, $s = 5$, and $r = 10$. **(b)** In this model, the online inpainter still uses memory of the last frames it inpainted, but it also receives information from the inpaintings of the second model. This refining inpainter performs a slower but better inpainting as it can work directly on windows. If tuned correctly, both models process the video at the same speed, so that the refined memory is always relevant to the online inpainter. Following Equation equation 4, we have here $f = 18$, $t = 14$ (for this example), $s = s' = 3$, and $r' = 10$ (tunable parameters).

This improved model addresses the issue by combining both worlds of high FPS and good quality with two inpainting modules working side by side (Figure 3 (b)). The first one, called the Online Inpainter, is very similar to the first model as it inpaints one frame at a time reusing results from previous inpaintings. As it computes attention linearly and not quadratically in the transformers, this inpainter displays real-time performances. The second module is called the Refining Inpainter and it works on already gone frames. Like the original approach, it inpaints frames with small windows to get the best inpainting quality allowed by the model. However, these inpainted frames are not meant to be displayed as a result have been already shown, and the live video is now further away. Instead, these inpaintings are only created to be used by the Online Inpainter, whose quality will increase as a result. The underlying idea is that if fine-tuned properly, both models will progress approximately at the same speed, making the refined inpaintings relevant for the Online Inpainter. It will therefore keep its real-time inpainting speed, but will be provided with much better previous inpaintings, increasing its own inpainting quality. For maximum speed, we implement this model using two GPUs, each of them carrying one of the inpainting modules. As detailed in Equation equation 4, we found the best input to be composed of neighboring frames from the Online Inpainter's memory as well as neighboring and reference frames from the Refined Inpainter's one.

$$\hat{Y}_f = \mathcal{M}'' \left( X_f, M_{f-s}^{f-1} \cup \bar{M}_{t-s'}^t \cup \bar{M}_{1,r'}^t \right). \tag{4}$$

The new model $\mathcal{M}''$ uses the same memory $M$ with span $s$ and now also the refined memory $\bar{M}$, with span $s'$ and sampling rate $r'$. $t$ is the index of the last inpainted frame available in this memory.

## 4 EXPERIMENTS AND RESULTS

### 4.1 BACKBONES

The models that we have implemented are online adaptations of state-of-the-art video inpainting transformers. There are three in total, and we will refer to them as "backbones". The first one is the Decoupled Spatial-Temporal Transformer (DSTT) (Liu et al., 2021b) which interweaves temporally-decoupled and spatially-decoupled transformers. The first ones focus on finding information in the same frame while the others try to find relevant content in other frames but at the same spot in the frame. The second model adapted in our work is the FuseFormer (Liu et al., 2021a), which uses traditional vision transformers on overlapping patches of the video. These patches are then blended together to enable a better reconstruction of the missing area. The last backbone is the End-to-End Framework for Flow-Guided Video Inpainting (E2FGVI) (Li et al., 2022), and it combines

Table 1: Results on the video reconstruction task for the DAVIS dataset.

| Backbone | Model | PSNR ↑ | SSIM ↑ | VFID ↓ | $E_{warp}$ ($\times 10^{-2}$) ↓ | FPS ↑ |
|---|---|---|---|---|---|---|
| DSTT | Offline | 31.25 | 0.959 | 0.137 | 0.152 | 19.2 |
| | + Online | 30.63 | 0.954 | 0.144 | 0.153 | 12.4 |
| | + Memory | 30.20 | 0.951 | 0.142 | 0.154 | 39.5 |
| | + Refined | 30.43 | 0.953 | 0.142 | 0.154 | 27.1 |
| FuseFormer | Offline | 31.74 | 0.962 | 0.126 | 0.152 | 10.8 |
| | + Online | 31.11 | 0.958 | 0.135 | 0.153 | 7.1 |
| | + Memory | 30.49 | 0.953 | 0.142 | 0.153 | 31.0 |
| | + Refined | 30.75 | 0.956 | 0.139 | 0.153 | 21.3 |
| E2FGVI | Offline | 32.08 | 0.964 | 0.117 | 0.148 | 8.8 |
| | + Online | 31.20 | 0.959 | 0.128 | 0.150 | 5.6 |
| | + Memory | 30.54 | 0.954 | 0.129 | 0.151 | 11.4 |
| | + Refined | 30.92 | 0.957 | 0.129 | 0.153 | 9.4 |

Table 2: Results on the video reconstruction task for the YouTube-VOS dataset.

| Backbone | Model | PSNR ↑ | SSIM ↑ | VFID ↓ | $E_{warp}$ ($\times 10^{-2}$) ↓ | FPS ↑ |
|---|---|---|---|---|---|---|
| DSTT | Offline | 31.98 | 0.960 | 0.054 | 0.099 | 17.9 |
| | + Online | 31.63 | 0.958 | 0.059 | 0.099 | 11.7 |
| | + Memory | 31.28 | 0.955 | 0.058 | 0.100 | 34.9 |
| | + Refined | 31.54 | 0.957 | 0.058 | 0.099 | 21.2 |
| FuseFormer | Offline | 32.34 | 0.962 | 0.053 | 0.098 | 10.3 |
| | + Online | 32.01 | 0.960 | 0.057 | 0.099 | 6.6 |
| | + Memory | 31.56 | 0.957 | 0.057 | 0.099 | 26.6 |
| | + Refined | 31.87 | 0.959 | 0.057 | 0.099 | 16.9 |
| E2FGVI | Offline | 32.65 | 0.963 | 0.049 | 0.096 | 8.5 |
| | + Online | 32.15 | 0.961 | 0.053 | 0.097 | 5.1 |
| | + Memory | 31.61 | 0.958 | 0.052 | 0.097 | 10.2 |
| | + Refined | 31.98 | 0.960 | 0.052 | 0.099 | 8.3 |

focal vision transformers with a flow-based approach. Bidirectional flows are computed to propagate content to the border of the missing region while the remaining part is completed by the transformers.

## 4.2 DATASETS

We compare the different approaches on two widely used datasets for video inpainting. They are composed of small videos with rather simple motions, more complex videos still being out of the scope of current state-of-the-art models. YouTube-VOS (Xu et al., 2018) is composed of 4519 videos of about 150 frames, while DAVIS (Perazzi et al., 2016; Pont-Tuset et al., 2017) contains 150 videos of about 120 frames. As we aim to adapt already existing models with our framework, we do not conduct supplementary training and reuse the weights of the pre-trained models. Therefore, we only use the test sets of these datasets. Following previous works (Liu et al., 2021a;b), we first quantitatively evaluate these videos on a reconstruction task using the same stationary masks as FuseFormer (Liu et al., 2021a). We then visualize some object removal inpaintings, on the DAVIS dataset. The lack of ground truth for this last task makes a quantitative analysis impossible.

## 4.3 QUANTITATIVE RESULTS

To assess quantitatively our models, we use 4 metrics frequently used in previous works (Zeng et al., 2020): PSNR, SSIM (Wang et al., 2004), VFID (Wang et al., 2018) and $E_{warp}$ (Lai et al., 2018). More specifically, PSNR (Peak Signal to Noise Ratio) and SSIM (Structural Similarity) are widely used techniques for assessing the quality of reconstruction of an image compared to its original version. VFID (Video-based Fréchet Inception Distance) assesses the visual quality of the whole video by calculating its closeness with natural videos using a pre-trained I3D model (Carreira & Zisserman, 2017). $E_{warp}$ (Flow warping error) is a more recently used metric of temporal consistency and can vary from a publication to another as it depends on the flow model used. The frame rate is

measured on RTX 2080 Ti GPUs. Except for the refined memory-based model, which runs on two GPUs (one for each inpainter), all models run on one GPU only.

As the ground truth video is required for all these metrics, only the reconstruction task is possible here: stationary masks are applied to normal videos, and the model tries to reconstruct the original video as close as possible. We compare our three online models, as well as the original offline model, using inputs of similar sizes (same number of neighboring and reference frames). The results for this task are reported in Table 1 for DAVIS and Table 2 for YouTube-VOS.

The first thing we can observe in both tables is the lead of the offline model, able to use information further away in the video to inpaint a given frame. This shows the need to establish a proper baseline as online video inpainting proves to be substantially harder than regular video inpainting. Then, we observe recurrent trends among the online models, regardless of the backbone. First, the basic online model is the best in terms of quality but also the slower. This is because every frame is inpainted by processing a whole window of previous frames. This gives a great inpainting, however only the inpainting of the last frame can really be used, making it too slow.

Then, using our memory-based technique, we observe a significant gain in the frame rate, with a factor of 2 for E2FGVI and even a factor of 3 for DSTT and FuseFormer. This is because after each transformer, the inpainting results are saved in memory so that it is not necessary to recompute them later for a future frame. This causes, however, a loss in quality as the frames are not really inpainted together anymore: one frame is inpainted at a time, and even if it uses information from past frames, it is not the same as inpainting them all together at once.

The last model, using the memory approach with refinement, tries to combine the best of both worlds. It still uses an inpainting memory to skip extra calculations, so the frame rate is still greatly improved compared to the baseline model (with a factor of 2 on DSTT and FuseFormer). And because it also reinpaints frames together on the side to reuse these results, it can also reduce most of the loss observed with memory only. We should point out that it is not unfair to use two GPUs here, because the Online Inpainter alone is fast enough to inpaint the frames in real time, something a slower model cannot do, even with more GPUs (only one can inpaint a given frame).

While the quality gains and losses are similar compared to other backbones, our models coupled with E2FGVI show less pronounced results on the frame rate. The reason is that our memory approach saves a lot of time only on the attention calculations inside the transformers, as shown in Figure 2. For DSTT and FuseFormer, these operations represent an overwhelming part of the time spent, while this is less true for E2FGVI: the calculation of flows takes time, as well as the calculation of the windows for the focal attention. These parts may surely be optimized as well, but this lies outside the scope of our framework at the moment.

### 4.4 MODELS COMPARISON

It can be difficult to fairly compare two models if we include the frame rate as a criterion as there is a manifest trade-off between speed and quality when changing the size of the input window: the more frames provided as context, the better the inpainting but also slower. To take that into account, we evaluated them using different input sizes and reported the results in a quality/speed plot, shown in Figure 4. The PSNR is chosen as a measure of the inpainting quality here.

With that plot, we can observe that the different models belong to different domains regardless of the input size: the baseline online model will never be as fast as the memory-based one, even with really small inputs, and vice-versa for the quality. One model will then be preferable to another, depending on the need of the user. As expected, the baseline model is the best for quality, the memory-based gives the best frame rate, and the refined memory-based fits in the middle. We can nevertheless note that for similar PSNR values, the refined model always shows higher frame rates compared to the baseline model, and is consistently above 20 FPS for both DSTT and FuseFormer backbones. As already mentioned, the results for E2FGVI are less outstanding and fail to meet the real-time speed.

### 4.5 ABLATION STUDY

In Table 3 we propose an ablation study of the previously chosen model to see the importance of each component in the input. As recalled by Equation equation 4, the refined memory-based

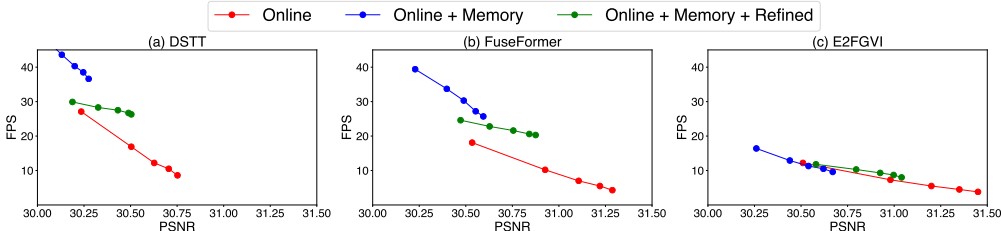

Figure 4: PSNR/FPS operating points on each backbone, using different input sizes.

Table 3: Ablation study on the input of our proposed model

| Neighboring Refined | Neighboring Online | Reference Refined | PSNR | SSIM | FPS |
|:---:|:---:|:---:|---|---|---|
| ✓ | | | 29.94 | 0.948 | 29.5 |
| ✓ | ✓ | | 30.40 | 0.952 | 23.3 |
| ✓ | | ✓ | 30.58 | 0.954 | 23.0 |
| ✓ | ✓ | ✓ | 30.87 | 0.956 | 19.7 |

model takes as input a) neighboring frames from the online memory, b) neighboring frames from the refined memory, and c) reference frames from the refined memory. As confirmed by the ablation, all components are helpful to inpaint the video. Reference frames seem especially important and cannot be simply replaced by more neighboring frames. Moreover, neighboring frames from the online memory are also really useful, certainly because they are really close to the current frame. More details about the parameter tuning of the model can be found in Annex of this paper.

### 4.6 QUALITATIVE VISUALIZATION

As one of the most promising in our study, we pick the refined memory-based model coupled with FuseFormer, having both good quality and frame rate. We perform an object removal task on the DAVIS dataset and report visual results for some of the videos in Figure 5.

## 5 DISCUSSION AND CONCLUSION

In this work, we presented a reliable method to adapt existing inpainting transformers to online and real-time standards while tempering quality loss. Some issues still remain, such as cases when little to no information is available to predict content, for example for the very first frames of the video as discussed in Appendix **??**. Moreover, our framework does not work yet with non-transformer models, limiting the adaptation of future techniques.

Nonetheless, we hope that by proving that both quality and speed could be met at the same time in video inpainting, we paved the way for more research in that direction. Eventually, stronger models will make online video inpainting accessible to the public, changing the way we produce live content.

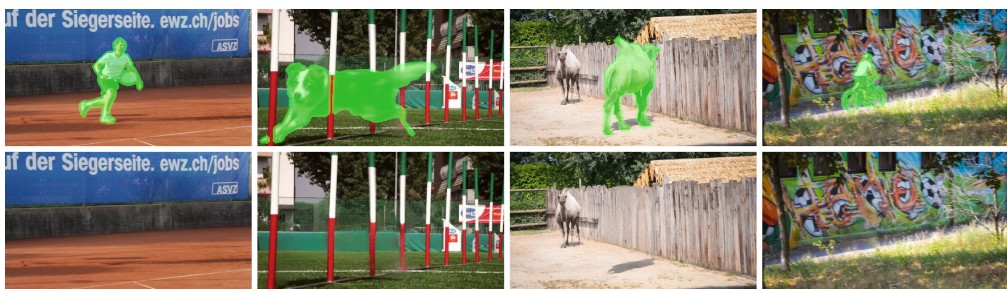

Figure 5: Visualization of some object removals performed at 20 FPS.

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
