# OpenReview forum: "Towards Online Real-Time Memory-based Video Inpainting Transformers"
_ICLR.cc/2023/Conference — Submitted to ICLR 2023_

### Official Review · Reviewer_ApZX · 2022-10-18

**Confidence:** 5
**Clarity, Quality, Novelty And Reproducibility:** Please find more details in the Weakn…
**Correctness:** 2
**Technical Novelty And Significance:** 2
**Empirical Novelty And Significance:** Not applicable
**Recommendation:** 3

**Strength And Weaknesses:**

### Strength
The task of real-time video inpainting is significant.
This paper is straightforward and easy to follow.

### Weaknesses

**Clarity**

The clarity of this paper needs grand improvements. Some important details are missing, for example,
1. the authors claimed ‘real-time’ in the Abstract but without providing the device information and video resolution.
2. In the caption of Figure 2, the authors should give a brief explanation of the ‘s’ when it first appears.
3. The authors should provide the memory cost for a comprehensive comparison of the experiments.
4. The authors didn’t provide video resolution in experiments.

**Novelty**
The core contribution of this paper is to use two inpainters to improve inpainting results. This solution is quite more like an engineering solution and the novelty is limited.

**Correctness**
1. The authors only conduct experiments on three transformer networks. However, some CNN-based video inpainting approaches have achieved a better trade-off between speed and quality, e.g., VINet (Kim et al., 2019) and LGTSM (Chang et al., 2019a). The authors should report and compare the proposed approach with them.
2. As mentioned in Section 4.2, the authors aim to adapt already existing models with the proposed framework. We found such a framework does not require a transformer-based approach. We suggest the authors adapt some CNN-based approaches with this two-inpainters framework to see the benefits.
3. How to ensure the inpainting memory in Figure 3 will get better? If a video frame with a large missing region results in a bad video inpainting frame in the memory. Is it better to discard this frame to avoid error accumulation rather than keeping refine it by a refining inpainter?
4. It’s not fair to report as ‘real-time’ on a 2080 Ti GPU, since this device is more powerful than most customer devices (e.g., phones, laptops).
5. Since the target is video inpainting, we suggest the authors provide a video demo for evaluation.


**Summary Of The Paper:**

This paper aims to design a real-time video inpainting framework. Specifically, the authors propose to use two inpainters, i.e. an online inpainter and a refining inpainter, to achieve a better trade-off between speed and video quality. This paper shows experimental results on DAVIS and Youtube-VOS based on three baselines, i.e., DSTT, FuseFormer and E2FGVI.

**Summary Of The Review:**

This paper targets at real-time video inpaitning, which is a significant and interesting topic. However, the novelty of the proposed framework is limited (seems like a combination of existing video transformer networks), and the experiments are not convincing (see more details in Weaknesses). Therefore, I tend to reject this paper. If all the concerns in Weaknesses can be well-addressed, I will raise my rating.

---

### Official Review · Reviewer_WiHT · 2022-10-24

**Confidence:** 3
**Correctness:** 3
**Technical Novelty And Significance:** 3
**Empirical Novelty And Significance:** Not applicable
**Recommendation:** 6

**Clarity, Quality, Novelty And Reproducibility:**

The writing of this work is good, and it is easy to follow. And the authors are suggested to release the code.

**Strength And Weaknesses:**

Strengths:
1.	 This method presents an online, memory, and refined video inpainting method.
2.	 Experimental results show the effectiveness of the developed method.
Weaknesses:
1.	The motivations are litter confused. As we all know, the memory mechanism and the transformer mechanism are time-consuming, but this work aims to achieve a real-time video inpainting performance with such two mechanisms.
2.	In the Introduction section, the authors are suggested to summarize the technical novelties of this method.
3.	It seems that the transformer mechanism has been widely used for video restoration. The authors should compare the developed method with them.
4.	It is possible to extend the developed online, real-time memory-based transformers in this work for other video restoration tasks?
5.	In Table 1 and Table 2, the authors are suggested to compare the model size of different methods.


**Summary Of The Paper:**

This work presents a video inpainting method based on transformers by memorizing and refining redundant computations while obtaining a decent inpainting quality. Experimental results show that the developed method can achieve online results with 20 frames per second.

**Summary Of The Review:**

This work has its merits. The motivations of this work are clearly presented, and the authors also present methods to address the target online real-time and memory-based video inpainting transformers. But this work also has several issues to be addressed; please refer to the weaknesses.

---

### Official Review · Reviewer_g9oW · 2022-10-24

**Confidence:** 5
**Correctness:** 4
**Technical Novelty And Significance:** 2
**Empirical Novelty And Significance:** 2
**Recommendation:** 6

**Clarity, Quality, Novelty And Reproducibility:**

The proposed method is of good practical value. However, the technical novelty may not be good enough.
I think more visual results are need to better investigate this method. Since offline version is definitely better, I think other metrics would be better to prove this method's effectiveness.

The paper writing and figures are clear and easy to understand.

**Details Of Ethics Concerns:**

No ethics concerns

**Strength And Weaknesses:**

Strengths
1. The proposed problem is of great practical value, which have long been ignored in previous methods.
2. The proposed memory and updating mechanics are very reasonable and practical.

Weaknesses
1. The main concern is the novelty part. Althought it is very important and very effective, the proposed memory, calculation order and memory refinement steps seem to be technical details, rather than novel contributions.
Maybe a new network design plus above mentioned memory mechanics would be a better and complete version, which can better highlight the value of above proposed modules.

2. In Table 1 and 2, since the proposed method is definitely worse than offline version, it is hard to evaluate quantitative results. How to more convincably prove its effectiveness ?

3. As a video processing task, and quantitative results cannot easily prove the method's effectiveness, I think more visual results are needed. However, only one is given in Fig. 5.

**Summary Of The Paper:**

This paper proposes a new method for real time video inpainting. More spefically, the authors base their method on recent inpainting models and use memory and re-organize calculation order to make them more efficient.

**Summary Of The Review:**

This paper proposes a new method for real time video inpainting. It addresses practical issues which preventing a real time usage and propose to use memory and refinement steps to accelerate recent inpainting models.

The proposed method is technically sound but seems lack of enough novelty.
And the effectiveness has not been thoroughly explored: visual results and other quantitative metrics would be better.

---

### Official Review · Reviewer_6qxK · 2022-10-25

**Confidence:** 3
**Clarity, Quality, Novelty And Reproducibility:** The paper is well written, more in th…
**Correctness:** 3
**Technical Novelty And Significance:** 2
**Empirical Novelty And Significance:** 4
**Recommendation:** 3

**Strength And Weaknesses:**

Strength:
- The paper is well-written, and the models are well evaluated both quantitatively and qualitatively.
- The idea of reusing precomputed frames in the attention layer to reduce the running time is innovative.
- The proposed two-modules pipeline can further balance the speed and quality.

Weaknesses:
- Online video inpainting is not a well-defined task. Video inpainting requires the mask inputs, while in this work, all the masks are not predicted but form the ground truth. Then it is not proper to call it online video inpainting, since the masks are not generated along with the new frames. Ground truth masks are practical for offline video inpainting since it is a post-processing task. What is the real user cases of online video inpainting? Though interesting, the reviewer is still confused about the values of the task.
- It seems the proposed method is designed for transformer-based / attention-based video inpainting work. However, for efficiency, flow-based or warping-based video inpainting may be more valuable. E2FGVI is time-consuming in computing the optical flow, making the proposed framework useless. The reviewer is curious about how those warping-based methods work in speed and quality, and whether we can turn to optimize the speed of convolutional-based model / optical-flow / homography-based model if we really want an online video inpainting model. The discussions related to it are not sufficient in this paper.
- In Table 1 and 2, why the online model can be slower than offline model? Are they using the same window size?
- For the refined model, is the window size smaller than the offline or baseline model? Why not display the results of the second refinement module directly? It's not very clear why we still need the first pipeline to save time given we already have an equal-fast module producing better results. Could we compare the results of refinement module only with the combined version?
- Figure 3 is not clear to parse. Different colors indicate the frame categories but very confusing.
- The reviewer cannot easily agree with the authors that using 2GPUs is a fair comparison. One can always use parallelization tricks to make the computation of attention more efficient. The metrics the authors need to add is the computation complexity, but not only the FPS. Given different input sizes and different GPU types, frame rate is less meaningful. For practical usage, it's not that common to deploy an online video inpainting model to a machine with 2 high-performing GPUs.
- The overall paper looks more like a technical report for engineering tricks, but not a paper discussing learning methods.


**Summary Of The Paper:**

The authors proposed an online video inpainting framework by using a special caching method and a parallel model to balance the quality and speed. Extensive experiments are conducted to evaluate the efficiency and quality, and the proposed method outperform the baselines when being applied to online cases.

**Summary Of The Review:**

The main concerns are,
- The task is interesting but not practical. Without online mask generation, online video inpainting is meaningless.
- The discussions of different types of video inpainting is not sufficient. The proposed methods can only be used to attention-based models. Optimization on the optical-flow / convolution / warping-based methods may be easier for improving the efficiency and quality at the same time. The authors may not be required to optimize them in this paper, but at least should compare the results in a similar way to E2FGVI.
-  The evaluation may not be clear enough, and computation complexity should be reported. While comparing FPS, the input frame size should be reported.

---

### Decision · Program_Chairs · 2023-01-20

**Decision:**

Reject

**Justification For Why Not Higher Score:**

The reviewers are not positive and there is no rebuttal.

**Justification For Why Not Lower Score:**

N/A

**Metareview: Summary, Strengths And Weaknesses:**

This submission improves existing video inpainting methods to make them online (running at 20 frames per second). Reviewers like the idea, but are concerned about the validity of the problem formulation, the presentation, and the rigorousness of the evaluation. There is no rebuttal. The AC agrees on the concerns and recommends rejection.